# Reduction of non-typeable results using a plasmid oriented Lymfogranuloma venereum PCR for typing of *Chlamydia trachomatis* positive samples

Pieter Willem Smit[1,2], Akke Rosanne Cornelissen[1], Sylvia Maria Bruisten[1,3]*

1 Public Health Laboratory (GGD) Amsterdam, Department of infectious diseases, Amsterdam, The Netherlands, 2 Medical Microbiology Laboratory, Maasstad ziekenhuis, Rotterdam, The Netherlands, 3 Amsterdam UMC, University of Amsterdam, Amsterdam Infection & Immunity Institute (AI&II), Amsterdam, The Netherlands

* sbruisten@ggd.amsterdam.nl

## Abstract

### Objectives

Typing of *Chlamydia trachomatis* (CT) is traditionally performed by characterising the *ompA* gene, resulting in more than a dozen different genovars, A to L. Type L is associated with Lymphogranuloma venereum (LGV) and commonly screened for using PCR, targeting the chromosomal *pmpH* gene. We aimed to develop and validate a new CT/LGV plasmid-based typing assay targeting the *pgp3* gene, to increase sensitivity and thus reduce the number of non-typeable results.

### Methods

The new *pgp3* PCR assay using LNA probes to detect point mutations was analytically and prospectively validated in a routine diagnostic laboratory setting. For the analytical tests, quantified nucleotide constructs (gBlocks) were used to perform limit of detection analyses. Quality control panel samples from 2018 and 2019 for CT were also tested. For the clinical study patient samples which were collected in two months in 2018 were tested simultaneously using the *pmpH* PCR and the *pgp3* PCR.

### Results

Analytically, the assay proved to be 100% specific relative to the previously used LGV typing assay targeting the single copy *pmpH* gene but it was much more sensitive to detect non-LGV CT. In the quality control panel 2 nonLGV samples and 7 LGV samples were solely positive with the *pgp3* PCR and not with the *pmpH* PCR. None of the samples from analytical specificity panels were positive, indicating 100% specificity. In a prospective panel of 152 clinical samples, 142 (93%) were successfully typed with the *pgp3* PCR compared to 78% with the *pmpH* PCR. The *pgp3* PCR was fully concordant with the *pmpH* PCR to

**Data Availability Statement:** All relevant data is included in the paper and its supplementary files.

**Funding:** The Authors received no specific funding for this work.

**Competing interests:** The authors have declared that no competing interests exist.

identify all LGV subtypes and detected an increased number of clinical samples of non-LGV subtype.

## Conclusion

We developed and validated a sensitive and specific plasmid-based typing assay to discriminate LGV from non-LGV CT subtypes. This is useful in a clinical setting to quickly determine the optimal treatment for *Chlamydia trachomatis* infections.

## Introduction

*Chlamydia trachom*atis (CT) infection is worldwide one of the leading bacterial sexually transmitted infections (STI) [1,2]. Typing of CT is traditionally performed by characterising the *ompA* gene, resulting in more than a dozen different genovars, A to L [3]. In the last decades other more discriminating typing techniques were also developed, leading to even more CT genotypes, which could be clustered in specific lineages [4,5]. Ocular *ompA* types are A, B, Ba and C and are still found only in non-Western countries but sometimes also occur in urogenital infections [4,6–10]. The urogenital genovars B and D to K, infect cervical and vaginal sites in women and anal sites in men and can all also be detected in urine samples. So genovars B and Ba may be found both at ocular and urogenital locations [6,9] All of the ocular and urogenital genovars are non-invasive types, in contrast to the *ompA* type L, which is associated with invasive infection leading to Lymphogranuloma venereum (LGV) disease, with swollen lymph nodes and anal ulcers. HIV seropositivity is a strong risk factor for LGV [11] and these LGV infections are mostly found in men who have sex with men (MSM) but in rare cases also in women [6].

LGV testing is strongly recommended for MSM with anorectal chlamydia since LGV associated types need longer and more stringent treatment. If untreated or treated inadequately LGV infection may cause chronic or irreversible complications, including fistulas, strictures, genital elephantiasis, frozen pelvis, or infertility [11, 12]. An urogenital nonLGV chlamydia is treated with a single dose of 1 g of azithromycin [13]. According to the most recent European guideline, an LGV infection is treated longer and more stringent with doxycycline 100 mg, twice a day, orally for 21 days [14]. This same treatment is recommended also in asymptomatic patients and contacts of LGV patients. If another regimen is used, a test of cure must be performed. Therefor it is very important to distinguish between LGV and non-LGV types. In many diagnostic laboratories, and also in our laboratory, a very sensitive screening assay is used to detect all Chlamydia types. In case of our laboratory this is the commercial Aptima Combo2 test which uses transcription mediated amplification (TMA) that detects RNA copies of the 23SrRNA gene [15]. Per bacterial cell hundreds to thousands of these RNA molecules are present leading to ultra-high sensitivity of the TMA. For subsequent typing to discriminate between LGV and nonLGV a diagnostic PCR was developed already in 2005 [16], targeting the *pmpH* gene. The sensitivity of this PCR was improved [17] but detection of this single copy *pmpH* gene was still at best 85% relative to the TMA screening assay. This leaves 15–20% of the CT positive samples non-typeable. In this study we now aimed to improve the LGV/non-LGV typing PCR by looking for suitable genes located on the CT plasmid, which was described to occur in 6 to 18 copies per bacterial cell [18–20], thus improving the sensitivity of the LGV typing PCR tenfold. The CT plasmid is 7.5 kb long and encodes 8 open reading frames (ORF), coding for several regulatory and virulence factors, being all plasmid glycoproteins (pGP) [21].

Indicative SNPs were assessed and finally pgp3-ORF5 was selected. The limit of detection, linearity and correctness in quality control panels were assessed. Subsequently we performed a clinical validation by testing diagnostic samples prospectively using both typing PCRs to compare sensitivity and specificity of the new plasmid gene PCR with the routinely used PCR targeting the single copy, chromosomal *pmpH* gene.

## Materials & methods

### Strategy of the study

Plasmid DNA sequences were obtained *in silico* from the BigsDB database and assessed for LGV differentiating SNPs [13]. The main goal was to reduce the proportion of non-typeable sample results. In addition, the acceptability criteria were to reach at least >95% sensitivity and specificity relative to the existing typing PCR, targeting the chromosomal *pmpH* gene. For the analytical validation quantified nucleotide constructs (gBlocks) and quality control samples were used. For the clinical validation diagnostic patient samples were tested simultaneously using the routine *pmpH* PCR and the plasmid gene PCR. Further details on the analytical and clinical part are described below.

**Nucleic acid extraction and PCR.** Nucleic acid (NA) extraction was performed as described previously [22]. In brief, isopropanol precipitation was applied to extract NA from 200 μL transport medium (Hologic, San Diego, USA) and the pellet was dissolved in 50 μL Tris-EDTA buffer pH 8.0. All CT positive samples were further tested with the *pmpH* LGV real time PCR [17]. Briefly, the real time PCR was performed in 20 μL, containing Platinum Quantitative PCR SuperMix-UDG (Invitrogen, Breda, the Netherlands), 2 μL of DNA solution, 4.3 mM MgCl$_2$, 0.40 μM of primer F3_LGV, 0.39 μM of primer F4_nonLGV and 0.92 μM of primer R2_ LGV/nonLGV, 0.15 μM of probe LGVtotP and 0.21 μM of probe P4_nonLGV (Table 1). Cycling conditions for the real-time PCR were: uracil DNA glycosylase step at 50˚C for 2 minutes and denaturation at 95˚C for 2 minutes, followed by 45 cycles of 15 seconds at 95˚C and 1 minute at 60˚C.

The *pgp3* real time PCR was performed in 20 μL, containing Quantinova Pathogen (QNP) master mix (Qiagen, Germany), 5 μL of DNA solution, 0.25 μM of forward and reverse primer, 0.125 μM of LGV probe and nonLGV probe with locked nucleic acids (LNA) for specificity to detect a point mutation. LNA probes and primers were conceived and ordered from IDT diagnostics (Integrated DNA technologies, Leuven, Belgium) and are shown in Table 1. Cycling conditions were identical to the *pmpH* PCR. All tests were performed on a Rotor-GeneQ thermocycler (Qiagen, Hilden, Germany).

**Table 1. Primer sequences used in this study.** Locked nucleic acids are indicated with '+'.

| LGV/nonLGV target | Primer/probe | Sequentie 5' —> 3' |
|---|---|---|
| *pmpH* core genome | F3_LGV | CTACTGTGCCAACCTCATCAT |
| | F4_nonLGV | CTATTGTGCCAGCATCGACTC |
| | R2_LGV/nonLGV | GACCCTTTCCGAGCATCA |
| | P4_nonLGV | Hex-AAAGAGCTTGAAGCAGCAGGAGC-BHQ2 |
| | LGVtotP | 6-FAM-CTTGCTCCAACAGT-MGB |
| PGP3_ORF5 | Forward_PGP3 | TTATTGCATCAAGAATGGAAGG |
| | Reverse_PGP3 | GCCTGATGAGTATCCATAACTA |
| | LGV probe_PGP3 | 56-FAM/CCC+T+A+CGC+G+AT/3IABkFQ |
| | nonLGV probe_PGP3 | 5HEX/CCC+T+G+CGCGA/3IABkFQ |

**Analytical validation.** The *pgp3* PCR efficiency and limit of detection was determined using spiked Tris-EDTA buffer pH 8.0 with the specific gBlocks (Integrated DNA technologies, Coralville, USA) with known concentration of CT DNA. A 10-fold dilution range was used in triplicate to determine the efficiency. To determine the 95% limit of detection (LOD95), a dilution series was prepared with factor 5 between the dilutions. Each dilution was tested 8 times.

Two proficiency panels (2018 and 2019) from Quality Control for Molecular Diagnostics (QCMD) were tested by the TMA and the two LGV/nonLGV PCR assays. Both panels contained CT and LGV positives and for each panel 9 out of 10 original samples were still available for this study. In addition, a CT panel was used to test the sensitivity for a diverse set of urogenital genovar types, D to K [23]. In addition, a specificity panel consisting of 48 microbial species, including several *Chlamydia* non-trachoma species, was used to assess the specificity of the newly developed assay.

**Prospective clinical study.** In September and October 2018 we included all routine samples which were sent to be tested for CT from patients attending the STI Public Health service of Amsterdam. The samples were screened for CT by amplification with the AC2 assay (Aptima Combo test, Hologic, USA) which tests concomitantly for CT and *Neisseria gonorrhoeae*. Anal swabs, ulcer swabs or urine samples from men were subsequently typed for LGV. Positive CT samples were genotyped using the validated *pmpH* polymerase chain reaction to differentiate LGV and non-LGV type infections [8].

**Ethical clearance.** This study was a methodological assessment to improve our diagnostic services. For this type of study the Medical Research Involving Human Subjects Act (WMO) does not apply, according to the Ethics of the Dutch law as declared by the Medical Ethics Review Committee AMC W19_496. #20.014.

## Results

### Test development

Full genome sequences of 157 *Chlamydia trachomatis* (CT) strains, including plasmid genes, were collected from the BIGsDB database (accessed in June 2017) [24]. ORF coding for the eight plasmid genes, *pgp1* to *pgp*8, were analyzed for single nucleotide polymorphisms (SNP's) differentiating LGV and non-LGV. These were detected in *pgp1* (ORF3) and *pgp3* (ORF5) with discriminating SNPs respectively on position 453 and 572 and conserved flanking sequences suitable for a forward and reverse primer. After initial technical validation, only the *pgp3* assay validation was continued since the *pgp1* assay proved to give low false-positive signals (not shown). In Fig 1 the alignment of the primers and probes relative to LGV and nonLGV consensus sequences is shown as generated from the 157 CT sequences. There is one G to A (LGV associated) SNP at position 572 and the LGV probe has an extra T nucleotide.

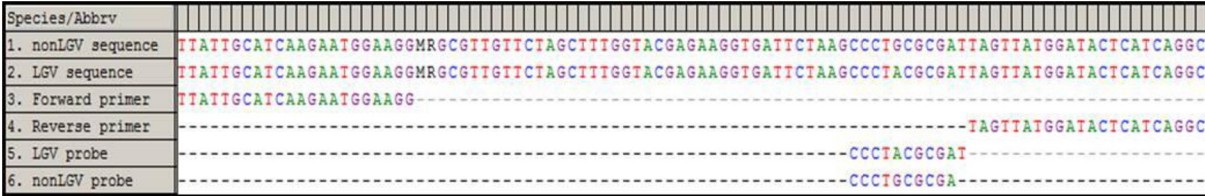

**Fig 1. Alignment of primers and probes for the CT-LGV typing PCR targeting the pgp3 PCR.** For the position of the LNA bases see Table 1.

## Technical specifications

The *pgp3* PCR efficiency was 1.87 for the LGV target and 1.93 for the non-LGV target, suggesting efficient amplification of ~2 (doubling of PCR product with each cycle). The 95% limit of detection (LOD95) was 0.95 copies/μL for LGV and 1.2 copies/μL for non-LGV.

## Specificity and sensitivity for different genovars

Genomic DNA from 48 non-*Chlamydia trachomatis* species, including *C. psittacci*, *C. caviae*, *C. muridarum* and *C. abortus* were tested. None were positive by the *pgp3* assay, indicating 100% specificity.

To assess the sensitivity of the *pgp3* assay, 23 DNA isolates from 8 different CT genovars D-K (all non-LGV) were tested as well. All were positive for non-LGV and negative for LGV. In addition, 18 quality control samples (9 from QCMD 2018 and 9 from QCMD 2019) were evaluated. The expected QCMD results were 2 nonLGV, 13 LGV and 3 negative samples. The TMA assay was 100% concordant for CT positivity with the expected QCMD results (S1 Table). The *pmpH* PCR assay had 47.4% concordance, with 0/2 nonLGV, 5/13 LGV and 3/3 negatives. In contrast the new *pgp3* PCR assay had 94.7% concordance, with 2/2 nonLGV, 12/13 LGV and 3/3 negatives (S1 Table).

## Prospective clinical validation

The new LNA *pgp3* PCR assay was performed in parallel to the routinely used *pmpH* PCR on a prospective panel of 152 routine diagnostic samples. There were 145 (95.4%) rectal swabs, 3 (2%) ulcer swabs, 2 (1.3%) urines and 2 (1.3%) samples of unknown origin.

Out of 152 samples, 119 (78.3%) were successfully typed with *pmpH* PCR and 142 (93.4%) samples with the new *pgp3* PCR assay. No LGV discrepancy was found between both assays (Table 2). The internal control was valid in all but one sample in these PCRs. So a substantial reduction in non-typeable results was observed, with only 6.6% for the *pgp3* PCR assay versus 21.7% not typeable for the *pmpH* PCR assay. All those that could be additionally typed were non-LGV samples.

Although both assays are not used quantitatively, the mean PCR Ct values were much lower for the *pgp3* assay compared to the *pmpH* target. For LGV positives the mean difference was 3.7 Ct (mean of Ct 28.8 for *pmpH* PCR and Ct 25.1 for *pgp3* PCR) and for non-LGV positives this was 3.2 Ct (mean of Ct 30.3 for *pmpH* PCR and Ct 27.1 for *pgp3*PCR). This translates to factor 13 (= $2^{3.7}$) and factor 6.4 (= $2^{3.2}$) increase in sensitivity for respectively the LGV and the nonLGV strains in patient samples.

## Discussion

In this study we developed and validated a new LGV typing assay, to replace or add to the commonly used *pmpH* PCR [16,17]. While the performance of the *pmpH* PCR to detect LGV types is mostly satisfactorily in the clinical setting, the performance to detect non-LGV was too

**Table 2. Prospective comparison of 152 clinical samples between the new *pgp3* PCR and the *pmpH* PCR to discriminate LGV from nonLGV types.**

| Total of 152 samples | *pmpH* PCR N (%) | *pgp3* PCR N (%) |
|---|---|---|
| LGV positive | 15 (9.9) | 15 (9.9) |
| Non-LGV positive | 101 (66.4) | 124 (81.6) |
| Double positive | 3 (2.0) | 3 (2.0) |
| Non-typeable | 33 (21.7) | 10 (6.6) |

low, resulting in 15% to 25% non-typeable outcomes in this and in previous studies [17,25]. The newly developed assay targets one of the CT plasmid genes, which is estimated to be up to18 fold more frequently present compared to genomic CT DNA [18] and for the highly prevalent genovars D, E and F it was reported to be even up to 30 plasmid copies [26]. Using the data from the 152 *Chlamydia trachomatis* (CT) positive samples that were typed in this study we noted a factor 6.4 increase for nonLGV and factor 13 for LGV, which may be translated to the presence of a mean of 6 plasmid copies in the nonLGV strains and 13 copies in the LGV strains. This is of course an inaccurate way of calculating plasmid copies and further studies are needed.

The new plasmid-based assay proved to have 100% sensitivity to detect LGV subtypes, while decreasing the non-typeable outcomes from 21.7% to 6.6%. No cross reactivity was observed for a wide range of organisms, including four other *Chlamydia* species, indicating the assay to be specific. One of these four was *C. muridarum* which is often used as a model for human *C. trachomatis* infection studies [27].

The chosen target region is expected to be more stable (see also below) than a previously used plasmid associated target in the commercial CT nucleic acid detection based assays from Roche and Abbott at that time [28,29]. A deletion in the target on the cryptic plasmid led to diminished detection of certain CT strains. This strain was named the 'Swedish variant' since it was predominantly found in Sweden [28,29]. Recently Borges et al (2019) reported that some LGV strains were missed by using only *ompA* typing of CT in Portugal. This was due to a CT strain in which part of the D/Da *ompA* gene was integrated by recombination in the L2b *ompA* gene in an L2 background CT genome. The variant LGV strain caused symptoms and was correctly diagnosed using another LGV typing test that targeted the *pmpH* gene [30]. We tested DNA from this recombinant *ompA* LGV strain and were able to detect <50 copies/μL, showing that the *pgp3* LGV PCR is also able to detect this recombinant strain.

In a previous study we reported that the CT genovar and its plasmid type are highly associated [13]. We expect that the *pgp3* gene on the plasmid is stable and thus suitable for diagnosis since it has been associated with virulence factors [20,21,24,27,31]. It has been reported that deficiency in the *pgp3* product, pGP3, leads to severely reduced virulence in *in vivo* mouse models using the closely related *C. muridarum* strain [21]. Plasmid free CT strains are rarely reported and *in vitro* mouse studies indicate that this type of strains are attenuated [19,27,32] so these may not pose a clinical problem in the human setting. More research is needed here.

In addition, our testing strategy is to first detect CT using a very sensitive screening assay (based on TMA) followed by typing CT positive samples derived from men with an ulcer or from an anal sample. According to the 2019 IUSTI guideline it is not recommended to include LGV subtype determination within standard STI screening assays, but only in case of a positive CT signal [14]. Presently, commercial assays are on the market that offer the possibility to type LGV strains as part of broad STI pathogen screening assay which leads to over-testing. Depending on local agreements with companies and costumers a commercial assay may be more expensive. Our LGV typing PCR is dedicated to only type those specimens that need to be typed according to the IUSTI recommendation [14], making it cost-efficient.

The fact that all samples that qualify for typing were already positive in the highly sensitive TMA screening assay minimises the possibility that certain CT strains will be missed. Diagnostic targets can however always be missed by single target assays, which was recently shown with the occurrence of mutations in the 23SrRNA gene, which caused missed infections in samples from Finland and Norway using the TMA AC2 assay from Hologic [33,34]. Therefor it may be a good testing strategy to perform a multiplex PCR, targeting both the *pgp3* and the *pmpH* genes in one reaction.

Because the plasmid-based LGV typing assay was a mean of 3.2 Ct values (for nonLGV strains) to 3.7 Ct (for LGV strains) lower, an increase in sensitivity is to be expected for both LGV and non-LGV subtypes. This can be seen in the QCMD panel results where also more LGV samples were detected using the *pgp3* PCR (S1 Table). Based on these results, and from our experience with more than two decades of routine testing for LGV, we speculate that LGV is mostly present in a higher bacterial load in clinical samples compared to non-LGV subtypes, so these LGV types were probably seldomly missed with the *pmp*H based assay.

To conclude, we developed a sensitive and specific plasmid-based typing PCR assay to discriminate LGV from non-LGV subtypes and to detect double LGV plus non-LGV infections. Further evaluations at other laboratories are recommended to confirm our findings.

## Supporting information

**S1 Table. QCMD 2018 and 2019 results.**
(XLSX)

**S1 File.**
(PDF)

## Acknowledgments

We thank Bart Versteeg for providing his data on full genomes of *Chlamydia trachomatis* including plasmid sequences. Researchers from Integrated DNA technologies are acknowledged for contributing to conceiving the LNA probes. We thank Vitor Borges for providing us DNA of the recombinant *ompA* D/L2 chlamydia strain and Servaas Morré and Jolein Pleijster for providing *C. muridarum* DNA.

## Author Contributions

**Conceptualization:** Pieter Willem Smit, Sylvia Maria Bruisten.

**Data curation:** Pieter Willem Smit, Akke Rosanne Cornelissen.

**Investigation:** Akke Rosanne Cornelissen, Sylvia Maria Bruisten.

**Methodology:** Akke Rosanne Cornelissen.

**Supervision:** Sylvia Maria Bruisten.

**Writing – original draft:** Pieter Willem Smit.

**Writing – review & editing:** Sylvia Maria Bruisten.

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
