## [Decision Letter · Decision Letter 0]

26 Feb 2020

PONE-D-19-34266

Reduction of non-typeable results using a plasmid oriented Lymfogranuloma venereum PCR for typing of Chlamydia trachomatis positive samples.

PLOS ONE

Dear Dr. Bruisten,

Thank you for submitting your manuscript to PLOS ONE. After careful consideration, we feel that it has merit but does not fully meet PLOS ONE’s publication criteria as it currently stands. Therefore, we invite you to submit a revised version of the manuscript that addresses the points raised during the review process.

Please provide a point by point response response to the reviewers' comments with appropriate revisions.

We would appreciate receiving your revised manuscript by two months. To enhance the reproducibility of your results, we recommend that if applicable you deposit your laboratory protocols in protocols.io, where a protocol can be assigned its own identifier (DOI) such that it can be cited independently in the future. For instructions see: http://journals.plos.org/plosone/s/submission-guidelines#loc-laboratory-protocols

We look forward to receiving your revised manuscript.

Kind regards,

Deborah Dean, M.D., M.P.H.

Academic Editor

PLOS ONE

Journal Requirements:

"Funding was provided by the Public Health laboratory for this study, no additional resources

were obtained."

Reviewers' comments:

Reviewer's Responses to Questions

**Comments to the Author**

1. Is the manuscript technically sound, and do the data support the conclusions?

Reviewer #1: Yes

Reviewer #2: Yes

2. Has the statistical analysis been performed appropriately and rigorously? 

Reviewer #1: N/A

Reviewer #2: Yes

3. Have the authors made all data underlying the findings in their manuscript fully available?

Reviewer #1: Yes

Reviewer #2: Yes

4. Is the manuscript presented in an intelligible fashion and written in standard English?

Reviewer #1: Yes

Reviewer #2: Yes

5. Review Comments to the Author

Reviewer #1: The study performed by Pieter W. Smit and colleagues aimed at presenting a sensitive and specific plasmid-based typing PCR assay to discriminate LGV from non-LGV subtypes. In my opinion, it deserves to be published if a proper revision is performed.

Major points:

- I have no doubts that this is a very useful study, presenting a very useful tool, which focuses a major ongoing problem in Chlamydia trachomatis. However, the authors do not properly explain why this is so important! There is a lack of information (either in the Introduction or in the Discussion), regarding the importance of detecting the LGV cases, the epidemic of the LGV cases during the last 10-15 years, the sexual networks behind this, the demand of multiple countries for the obligatory notification of the LGV cases, the different duration of the treatment, etc, etc.

- Also, the paper is not presented in an attractive fashion. It is sometimes too vague, lacking the rationale behind the used approaches. The order by which the sub-sections of the Methods is presented is a perfect example of this. The sub-set of samples are somehow confusing. The paper should follow the basic structure: Background on Chlamydia and LGV, identification of the problem, aims of the study to solve the problem, and adopted strategy.

- I would definitely add a detailed schematic figure presenting the plasmid region with the SNPs differentiating LGV and non-LGV strains and the location of the primers/probes.

Minor points to be considered:

-

- Page 3, Introduction: Refª 4 is too specific. I suggest replace it by a more general one or include additional refs;

- Page 3, Introduction: Line 5 states that types “B, D to K, infect cervical and vaginal sites (…)”. The reader may become confused as in the previous sentence, it is stated “ocular types are A, B and C…” Please rephrase it;

- Page 3, Introduction: “The LGV associated types need longer and more stringent treatment”. I would describe the dissimilar treatments for these two types of genital infections (non-LGV vs LGV) in order to make it a more comprehensive / interesting paper. At some instances, the paper seems excessively straightforward.

- Page 3, Introduction, last paragraph: “The sensitivity of this PCR was improved (8) but detection of this single copy pmpH gene was still at best 85% relative to the commercial TMA Aptima Combo2 test, as a screening assay, which detects RNA copies of the 23SrRNA gene (9)”. I suggest rephrasing or altering some refs because it is really weird the way it is written. In fact, ref 8 is from 2010 and ref 9 is from 2005. Obviously, ref 9 has nothing to do with ref 8. I would expect to see a ref posterior of 2008 to support the statement of the 85% sensitivity.

- Page 3, Introduction, last paragraph: As the present study is focused on the use of the plasmid, I would strengthen the refs supporting the plasmid copy number. Besides ref 10, I would add: Ferreira et al 2013, https://doi.org/10.1016/j.micres.2013.02.001 , and Pickett et al 2005, https://doi.org/10.1099/mic.0.27625-0

- Page 3, Methods: I suggest a different organization of the methods. Usually we start by the “study population”, the “N”, the “rationale for the different approaches”. It does not make sense to start by the “DNA extraction” without explaining the strategy and “N”;

- Page 5, Analytical validation: “fivefold dilution series was performed in eightfold”. Not sure what this means;

- Page 5, Analytical validation: Do not use the abbreviation TMA as it was not used before in the text;

- Page 5, Analytical validation, last paragraph: Please state how many samples were enrolled in these panels (i.e., QCMD and the second panel for types D to K). The information is too vague here. I realized later that only in page 7 this information is included.

- Page 7, Technical specifications: Where do this come from…? How was it calculated? What was the “N” use to calculate the efficiency? What is the meaning of “1.87” efficiency?

- Page 9, Discussion: Please rephrase the second sentence;

- Page 9, Discussion, line 6: I would add here the two references suggested above;

- Page 9, Discussion, line 7: Ref 15 is from an Abstract of a poster. I have no idea if PLoSOne accepts such references. Also, I do not know if the authors have used accurate real-time-based absolute quantitative approaches as the ones used in ref 10 and also in Ferreira et al 2013. Plasmid numbers of 30 seem to contrast with at least 3 previous accurate studies…

- Page 9, Discussion: Please complete ref 18. Why the bioRxiv? I found out that the paper is published.

- Page 10, Discussion, first sentence: Why ref 13, here? Others are much more appropriate, such as: Song et al 2013 (doi: 10.1128/IAI.01305-12), Lehr S et al 2018 (doi: 10.1016/j.micinf.2018.02.007), Zhong G 2017 (doi: 10.1016/j.tim.2016.09.006).

- Page 10, Discussion: “Presently, commercial assays are on the market that offer the possibility to type LGV strains as part of broad STI pathogen screening assay which thus are not in line with the IUSTI recommendations”. Why would this constitute a problem…? Only if it increments the cost of the already existing diagnostic test, right? Are they more expensive than the ones that exist for the last decade? Is that the case? If yes, please state it clearly, because it is a strong argument. However, you must be sure of what you are saying.

- Page 10, Discussion: “…an increase in sensitivity was only observed for non-LGV subtypes…” This statement makes no sense. One cannot compare such sensitivities by using radically different denominators (15 LGV samples versus more than 100 non-LGV samples). Please rephrase it.

Reviewer #2: The manuscript titled “Reduction of non-typeable results using a plasmid oriented Lymfogranuloma venereum PCR for typing of Chlamydia trachomatis positive samples” by Smit et al was submitted to be published in PLoS ONE.

The work describes a new typing strategy in Chlamydia trachomatis based on real time PCR using a single nucleotide variation (SNV) in pgp3 gene, located in the cryptic plasmid. The authors found a higher sensitivity around 10-fold respect to the classical insertion/deletion in pmpH gene for non-LGV CT subtypes.

The work is well presented, and the results are interesting. I have a few questions for the authors. For instance,

Why they used 2mcL of DNA in pmpH approach and 5 mcL in pgp3. Could this difference have any impact in the sensitivity?

In the specificity test, Chlamydia muridanum must be included, because this specie is phylogenetically more related to C. trachomatis compared to C. caviae, C. abortus or C. psittacci.

If the authors found identical sensitivity for LGV using pmpH or pgp3 approach, could be suggested the typing based on pmpH gene in first step and those non-typeable cases analyzed using pgp3 gene?

The difference in Ct value is 2 (not 3 cycles as the authors are proposing) and consequently the sensitivity could be low to 10-fold.

The QCMD panels must be described, briefly. The section of result referred to sensitivity should be explained clearly.

could be commented the impact of plasmid free C.trachomatis strains?.

I would recommended to review the supplementary information; could you present this data in English language?

6. PLOS authors have the option to publish the peer review history of their article (what does this mean?). If published, this will include your full peer review and any attached files.

Reviewer #1: No

Reviewer #2: Yes: Juan Carlos Galan

---

## [Author Response · Author response to Decision Letter 0]

24 Apr 2020

To the Editor,

PlosOne

April 24, 2020

Cover letter regarding manuscript: 

“Reduction of non-typeable results using a plasmid oriented Lymfogranuloma venereum PCR for typing of Chlamydia trachomatis positive samples.”

Dear Prof. dr. Deborah Dean,

We are pleased to re-submit a manuscript on Chlamydia trachomatis (CT) typing which describes a new assay that allows sensitive discrimination of lymphogranuloma venereum (LGV) versus non-LGV strains. We are very greatful for the constructive comments of the reviewers. All points that were raised were answered to the best of our knowledge.

We sincerely hope that you will consider our manuscript to be suitable for publication in PLOS ONE.

On behalf of our co-authors,

Sincerely yours,

Dr Sylvia M. Bruisten, PhD (corresponding author)

Public health laboratory

Department of Infectious diseases, 

Public Health Service of Amsterdam (GGD Amsterdam), 

Nieuwe Achtergracht 100, 

1018 WT Amsterdam, 

the Netherlands

Phone: +31-20-5555376

E-mail: sbruisten@ggd.amsterdam.nl

5. Review Comments to the Author

Reviewer #1: The study performed by Pieter W. Smit and colleagues aimed at presenting a sensitive and specific plasmid-based typing PCR assay to discriminate LGV from non-LGV subtypes. In my opinion, it deserves to be published if a proper revision is performed.

AU: Thank you for giving us the opportunity to improve our manuscript to make it more attractive for readers of PlosOne. We appreciate your input and your help with using the right references.

Major points:

- I have no doubts that this is a very useful study, presenting a very useful tool, which focuses a major ongoing problem in Chlamydia trachomatis.

However, the authors do not properly explain why this is so important! There is a lack of information (either in the Introduction or in the Discussion), regarding the importance of detecting the LGV cases, the epidemic of the LGV cases during the last 10-15 years, the sexual networks behind this, the demand of multiple countries for the obligatory notification of the LGV cases, the different duration of the treatment, etc, etc.

AU: Thank you for this valuable comment. Indeed we agree that detecting LGV types is very important and also to put this in the context of the ongoing LGV epidemic. We previously tried to submit this paper to another Journal where only a limited number of words and references were allowed. When opting for PlosOne we accidentally omitted to adjust the manuscript to all of the PlosOne requirements and use the possibility to expand the background and discussion. We have now elaborated more over these different subjects and have also used more suitable references. 

To our defense: this is in essence a technical paper and we did not aim to write a review on LGV epidemics etc. Nevertheless, we have now rewritten parts of the Introduction and Discussion to strengthen our message that it is very important to sensitively type Chlamydia trachomatis positive samples for LGV. 

- Also, the paper is not presented in an attractive fashion. It is sometimes too vague, lacking the rationale behind the used approaches. The order by which the sub-sections of the Methods is presented is a perfect example of this. The sub-set of samples are somehow confusing. The paper should follow the basic structure: Background on Chlamydia and LGV, identification of the problem, aims of the study to solve the problem, and adopted strategy.

AU; We again fully agree with the reviewer that the paper can be much improved by following a more strict format. We adjusted the structure of the total manuscript and hope that it has improved, making parts of the text more comprehensive.

Large parts of the manuscript were rewritten, with the study aims in the Introduction and we added and adjusted subject headings in Materials and Methods, which now starts with the study strategy which includes acceptance criteria. Sample selection is subdivided for the analytical phase and the clinical phase of this validation study; it seems most logical for us to still mention these in the relevant paragraphs. 

- I would definitely add a detailed schematic figure presenting the plasmid region with the SNPs differentiating LGV and non-LGV strains and the location of the primers/probes.

AU: Thank you for this suggestion which we gladly follow. A new Figure 1 has been added which shows the location of primers and probes in the pgp3 gene and also shows the SNP which distinguishes the LGV from the nonLGV types, relative to 157 sequences of strains that were aligned earlier and were retrieve from GenBank (ref Versteeg et al, BMC Genomics, 2018).

Minor points to be considered:

-

- Page 3, Introduction: Refª 4 is too specific. I suggest replace it by a more general one or include additional refs;

AU: We agree and replaced it for a couple of more general references (Andersson et al Nature comm, 2016; Giffard et al PlosOne 2018, Holt et al, Methods Mol Biol. 2019 Joseph et al, Mol biol Evol 2012; Andersson et al, Nature communications, 2016, Giffard et al, 2018; Holt et al, Methods Mol Biol. 2019;2042:87-122).

- Page 3, Introduction: Line 5 states that types “B, D to K, infect cervical and vaginal sites (…)”. The reader may become confused as in the previous sentence, it is stated “ocular types are A, B and C…” Please rephrase it;

AU: We agree. However, we first want to mention that ocular types are A, B, Ba and C and then come to the urogenital types which are B to K. Indeed types B and Ba are found both at ocular and urogenital sites. We do not think this is confusing but still added another sentence to point this out:: ‘ So genovars B and Ba may be found both at ocular and urogenital locations (6,9).’

- Page 3, Introduction: “The LGV associated types need longer and more stringent treatment”. I would describe the dissimilar treatments for these two types of genital infections (non-LGV vs LGV) in order to make it a more comprehensive / interesting paper. At some instances, the paper seems excessively straightforward.

AU: Again we fully agree. Again, this short previous description is due to not realizing that for PlosOne we are not restricted to the number of words, nor to the number of references. We have now elaborated on this on page 3 and . :

“ LGV testing is strongly recommended for MSM with anorectal chlamydia sinceThe LGV associated types need longer and more stringent treatment. If untreated or treated inadequately LGV infection may cause chronic or irreversible complications, including fistulas, strictures, genital elephantiasis, frozen pelvis, or infertility (van der Bij, 2006). An urogenital nonLGV chlamydia is treated with a single dose of 1 g of azithromycin (Lanjouw et al, Int J STD Aids, 2016). According to the most recent European guideline, an LGV infection is treated much longer and more stringent with doxycycline 100 mg, twice a day, orally for 21 days (20= de Vries et al, J Eur acad, 2019). This same treatment is recommended also in asymptomatic patients and contacts of LGV patients. If another regimen is used, a test of cure must be performed.” 

- Page 3, Introduction, last paragraph: “The sensitivity of this PCR was improved (8) but detection of this single copy pmpH gene was still at best 85% relative to the commercial TMA Aptima Combo2 test, as a screening assay, which detects RNA copies of the 23SrRNA gene (9)”. I suggest rephrasing or altering some refs because it is really weird the way it is written. In fact, ref 8 is from 2010 and ref 9 is from 2005. Obviously, ref 9 has nothing to do with ref 8. I would expect to see a ref posterior of 2008 to support the statement of the 85% sensitivity.

AU: We agree that these two tests, the CT-typing targeting the pmpH gene (ref 8) and the much longer existing TMA assay (ref 9) refer to different entities. In itself our statement was correct but we have now rephrased this part to: 

“In many diagnostic laboratories, and also in our laboratory, a very sensitive screening assay is used to detect all Chlamydia types. In case of our laboratory this is the commercial Aptima Combo2 test which uses transcription mediated amplification (TMA) that detects RNA copies of the 23SrRNA gene (15). Per bacterial cell hundreds to thousands of these RNA molecules are present leading to ultra-high sensitivity of the TMA. For subsequent typing to discriminate between LGV and nonLGV a diagnostic PCR was developed already in 2005 (16), targeting the pmpH gene. The sensitivity of this PCR was improved (17) but detection of this single copy pmpH gene was still at best 85% relative to the TMA screening assay..” 

- Page 3, Introduction, last paragraph: As the present study is focused on the use of the plasmid, I would strengthen the refs supporting the plasmid copy number. Besides ref 10, I would add: Ferreira et al 2013, https://doi.org/10.1016/j.micres.2013.02.001 , and Pickett et al 2005, https://doi.org/10.1099/mic.0.27625-0

AU: Thank you for your valuable suggestions of these references. We have now incorporated those mentioned and also some additional ones. We rephrased the statement to: 

 “In this study we now aimed to improve the LGV/non-LGV typing PCR by looking for suitable genes located on the CT plasmid, which was described to occur in 6 to 18 copies per bacterial cell (10) (15, Pickett el al, 2005; Ferreira et al, 2013) , thus improving the sensitivity of the LGV typing PCR tenfold. The CT plasmid is 7.5 kb long and encodes 8 open reading frames (ORF), coding for several regulatory and virulence factors, being all plasmid glycoproteins (pGP) (Zhong, Trends Microbiol, 2017) . (Page 4, Introduction)

- Page 3, Methods: I suggest a different organization of the methods. Usually we start by the “study population”, the “N”, the “rationale for the different approaches”. It does not make sense to start by the “DNA extraction” without explaining the strategy and “N”;

AU: We partly agree, please see also our reply in the general comments above.

- Page 5, Analytical validation: “fivefold dilution series was performed in eightfold”. Not sure what this means;

AU: This means that a dilution series was prepared with factor 5 between the dilutions. Each dilution was tested 8 times. This was also adjusted in the manuscript:

“To determine the 95% limit of detection (LOD95), a dilution series was prepared with factor 5 between the dilutions. Each dilution was tested 8 times. (Page7 Methods)

- Page 5, Analytical validation: Do not use the abbreviation TMA as it was not used before in the text;

AU: We have now adjusted it to ‘Transcription mediated amplification (TMA)’. In the same line, we have now also written ‘QCMD’ in full in the first instance where it appeared (Page 5).

- Page 5, Analytical validation, last paragraph: Please state how many samples were enrolled in these panels (i.e., QCMD and the second panel for types D to K). The information is too vague here. I realized later that only in page 7 this information is included.

AU: We agree and have now added more details both on page 5 and also in Results (Page,9) where we now refer to a Supplementary Table 1. In this added table all data on results for TMA, pmpH PCR and pgp3 PCR are shown, next to the expected results from QCMD.

- Page 7, Technical specifications: Where do this come from…? How was it calculated? What was the “N” use to calculate the efficiency? What is the meaning of “1.87” efficiency?

AU: PCR efficiency is expressed from 0-2 as being the amplification efficiency of the PCR product. The best PCR doubles its product with each cycle (2), the worst doesn’t amply at all, and thus has efficiency of 0. This has been addressed in the manuscript;” The pgp3 PCR efficiency was 1.87 for the LGV target and 1.93 for the non-LGV target, suggesting efficient amplification of ~2 (doubling of PCR product with each cycle).” 

- Page 9, Discussion: Please rephrase the second sentence;

AU: This sentence is now rephrased as follows: While the performance of the pmpH PCR to detect LGV types is sensitive, the performance to detect non-LGV was too low, resulting in 15% to -25% non-typeable outcomes in this and in previous studies (8,14).

- Page 9, Discussion, line 6: I would add here the two references suggested above;

AU: Indeed we have now added these references as also mentioned above. 

- Page 9, Discussion, line 7: Ref 15 is from an Abstract of a poster. I have no idea if PLoSOne accepts such references. Also, I do not know if the authors have used accurate real-time-based absolute quantitative approaches as the ones used in ref 10 and also in Ferreira et al 2013. Plasmid numbers of 30 seem to contrast with at least 3 previous accurate studies…

AU: Using an abstract as reference seems to be allowed for PlosOne (the editor may know this) so we have kept this reference. We now added also other references:

 Pickett et al, Microbiol 2005; Song et al, Inf and Immun 2013; Lehr et al, microbes and Inf, 2018),. For our assay we quantitated by using known numbers of the gBlock constructs, please see result section.

With respect to the 30 plasmid copies in one bacterial cell: this is what the authors from that study claim. It seems that, depending on the Chlamydia lineage and on the virulence, the copy numbers of plasmids vary indeed between 6 to 30 copies. For our diagnostic assay, which is a qPCR that amplifies exponentially with factor 2 each cycle, this just means the following: if there is a ten-fold difference in input (so 10 plasmid copies per CT versus 1 chromosomal gene copy) the Ct value difference is 3.322 (= 2log(10)). In practice we measured a mean difference of 3.7 Ct values (=28.8 for pmpH – 25.1 for pgp3) for LGV strains which would be in agreement with a mean plasmid copy number of 13 (=2e3.7 for LGV types). For nonLGV types we found a mean difference of 3.2 Ct values (30.3 for pmpH – 27.1 for pgp3) leading to an estimated plasmid copy number of 6.4. So indeed, 30 plasmid copies is probably rare and 6 copies (for nonLGV) to 13 copies (for LGV) seems more realistic.

- Page 9, Discussion: Please complete ref 18. Why the bioRxiv? I found out that the paper is published.

AU: Indeed, when submitting to previous journals this paper was not published yet. This has now been updated (reference 30). 

- Page 10, Discussion, first sentence: Why ref 13, here? Others are much more appropriate, such as: Song et al 2013 (doi: 10.1128/IAI.01305-12), Lehr S et al 2018 (doi: 10.1016/j.micinf.2018.02.007), Zhong G 2017 (doi: 10.1016/j.tim.2016.09.006).

AU: Thank you again for pointing out these interesting publications to us. We have looked up these references and indeed included them all now. We keep ref 13 since we showed here that the CT core genome and its plasmid genes are related per CT type this has been better specified Since PlosOne has no restriction on the number of references we are pleased to be able to expand the reference list.

- Page 10, Discussion: “Presently, commercial assays are on the market that offer the possibility to type LGV strains as part of broad STI pathogen screening assay which thus are not in line with the IUSTI recommendations”. Why would this constitute a problem…? Only if it increments the cost of the already existing diagnostic test, right? Are they more expensive than the ones that exist for the last decade? Is that the case? If yes, please state it clearly, because it is a strong argument. However, you must be sure of what you are saying.

AU: Actually there is no problem with using commercial tests in itself but in the guideline is recommended to first screen for CT and, if positive to also perform additional testing by typing for LGV in case of anal samples in men. Many commercial assays do both tests (CT detection and LGV typing) simultaneously, leading to over-testing. This is not recommended in the guideline for obvious reasons (See: de Vries et al, J Eur acad, 2019). 

Indeed this in-house PCR is less expensive than most commercial assays in our setting. This can however not be a general argument, since it depends on many factors, such as cost of personal, deals with companies where the kits are purchased and of course also how much can be charged for performing the tests. Therefor we did not explicitly mention the cost argument. We have now adjusted this part (page 12, second paragraph). 

- Page 10, Discussion: “…an increase in sensitivity was only observed for non-LGV subtypes…” This statement makes no sense. One cannot compare such sensitivities by using radically different denominators (15 LGV samples versus more than 100 non-LGV samples). Please rephrase it.

AU: We agree and have rephrased this part to: 

Even though the plasmid-based LGV typing assay was a mean of 3.2 Ct values (for nonLGV strains) to 3.7 Ct (for LGV strains) lower, an increase in sensitivity is to be expected for both LGV and non-LGV subtypes. This can be seen in the QCMD panel results where also more LGV samples were detected using the pgp3 PCR (S1_Supplementary table 1).

Reviewer #2: The manuscript titled “Reduction of non-typeable results using a plasmid oriented Lymfogranuloma venereum PCR for typing of Chlamydia trachomatis positive samples” by Smit et al was submitted to be published in PLoS ONE.

The work describes a new typing strategy in Chlamydia trachomatis based on real time PCR using a single nucleotide variation (SNV) in pgp3 gene, located in the cryptic plasmid. The authors found a higher sensitivity around 10-fold respect to the classical insertion/deletion in pmpH gene for non-LGV CT subtypes.

The work is well presented, and the results are interesting. 

AU: Thank you.

I have a few questions for the authors. For instance,

Why they used 2mcL of DNA in pmpH approach and 5 mcL in pgp3. Could this difference have any impact in the sensitivity?

AU: In a PCR reaction, which is an exponential increase of amplimers, the original input does not formidably influence the Ct value. An input of 2µL versus 5µL is a factor 2.5 difference which will lead to a maximum difference in Ct value of 1.32 ( 21.32=2.5). In routine diagnostics there are always flaws, such as contaminants which (partly) inhibit a PCR. Therefor we define a difference in Ct values of <1 as ‘not relevant’. So also a difference in input of 2µL versus 5µL is really not much influencing the qualitative outcome: positive or negative. The choice for input volume is made on practical grounds, once it is used in combination with a certain mastermix or certain primer/probe sets.

In the specificity test, Chlamydia muridanum must be included, because this specie is phylogenetically more related to C. trachomatis compared to C. caviae, C. abortus or C. psittacci.

AU: We indeed see in the literature that C. muridarum is often mentioned as a good mouse model for human pathogenesis with C. trachomatis. Besides testing C. muridarum DNA, we also assessed the performance of the PCR using PrimeReport. Both showed that the pgp3 PCR is unable to detect in silico and in-vitro C. muridarum. We adjusted this both in Methods and results.

If the authors found identical sensitivity for LGV using pmpH or pgp3 approach, could be suggested the typing based on pmpH gene in first step and those non-typeable cases analyzed using pgp3 gene?

AU: This may indeed be a strategy, but then labs would have to perform multiple successive testing steps and that would delay the test result. If the aim is that no strains are missed in case one of the genes would mutate, it would be better to combine both targets (so perform a multiplex test) and then declare the result as positive if one of both targets is positive. But actually it is smarter to first use the TMA assay (since it is more sensitive) and then use the pgp3 assay, since it is more sensitive than the pmpH PCR. If in doubt (for example that plasmids may be lacking or that mutations had arisen in the primers or probe for the pgp3 gene, then use the pmpH gene PCR. as confirmatory assay. We have now added the strategy to multiplex both genes in the text (Discussion, page 12: Therefor it may be a good testing strategy to perform a multiplex PCR, targeting both the pgp3 and the pmpH genes in one reaction.

The difference in Ct value is 2 (not 3 cycles as the authors are proposing) and consequently the sensitivity could be low to 10-fold.

AU: we do not really get the point here. If we look at our Ct values these actually do show a mean difference of 3.2 Ct values (for nonLGV strains) and 3.7 Ct (for LGV strains. See also our reply to comments of reviewer 1.

The QCMD panels must be described, briefly. The section of result referred to sensitivity should be explained clearly.

AU: Thank you for this comment. We agree that the QCMD panels were not adequately described, as was also remarked by the other reviewer. We have now made adjustments in Methods and Results and added a Supplementary Table 1 in which all QCMD results are shown in detail.

could be commented the impact of plasmid free C. trachomatis strains?

AU: This is an important point, thank you for bringing it up. Actually an increasing number of publications show that several plasmid ORFs code for virulence factors (refs…). And truly plasmid free CT bacteria seem to be attenuated. We have now commented on this (Page…):

Plasmid free CT strains are rarely reported and in vitro/mouse studies indicate that this type of strains are attenuated (Song et al, Inf and Immun 2013; Lehr et al, microbes and Inf, 2018), so these may not pose a clinical problem in the human setting. More research is needed here.

I would recommended to review the supplementary information; could you present this data in English language?

AU: We do not know if we correctly understand this comment. In the original manuscript there was no supplementary data. Only the ethical statement which was supplied as demanded by the editor, was both in Dutch and in English. The Dutch version was only given to show the literal text. 

In the present manuscript a Supplementary Table 1 was added showing the QCMD data; this table is in English only.

---

## [Decision Letter · Decision Letter 1]

18 May 2020

Reduction of non-typeable results using a plasmid oriented Lymfogranuloma venereum PCR for typing of Chlamydia trachomatis positive samples.

PONE-D-19-34266R1

Dear Dr. Bruisten,

We are pleased to inform you that your manuscript has been judged scientifically suitable for publication and will be formally accepted for publication once it complies with all outstanding technical requirements.

With kind regards,

Deborah Dean, M.D., M.P.H.

Academic Editor

PLOS ONE

Additional Editor Comments (optional):

Reviewers' comments:

Reviewer's Responses to Questions

**Comments to the Author**

1. If the authors have adequately addressed your comments raised in a previous round of review and you feel that this manuscript is now acceptable for publication, you may indicate that here to bypass the “Comments to the Author” section, enter your conflict of interest statement in the “Confidential to Editor” section, and submit your "Accept" recommendation.

Reviewer #1: All comments have been addressed

Reviewer #2: All comments have been addressed

2. Is the manuscript technically sound, and do the data support the conclusions?

Reviewer #1: Yes

Reviewer #2: Yes

3. Has the statistical analysis been performed appropriately and rigorously? 

Reviewer #1: N/A

Reviewer #2: N/A

4. Have the authors made all data underlying the findings in their manuscript fully available?

Reviewer #1: Yes

Reviewer #2: Yes

5. Is the manuscript presented in an intelligible fashion and written in standard English?

Reviewer #1: Yes

Reviewer #2: Yes

6. Review Comments to the Author

Reviewer #1: (No Response)

Reviewer #2: The new version has greatly improved respect to first draft. The manuscript is now easy to read and the too vague aspects have been clarified. The authors have given response to all reviewer's questions. In fact, they have introduced in main text many of the recommendations suggested by reviewers.

The current version help us to understand better the impact and improvement in the LGV diagnosis.

7. PLOS authors have the option to publish the peer review history of their article (what does this mean?). If published, this will include your full peer review and any attached files.

Reviewer #1: No

Reviewer #2: Yes: Juan Carlos Galán Microbiology Department, Hospital Ramón y Cajal, Madrid, Spain; CIBER in Epidemiology and Public Health, Spain.

---

## [Editor Report · Acceptance letter]

20 May 2020

PONE-D-19-34266R1 

Reduction of non-typeable results using a plasmid oriented Lymfogranuloma venereum PCR for typing of *Chlamydia trachomatis* positive samples. 

Dear Dr. Bruisten:

I am pleased to inform you that your manuscript has been deemed suitable for publication in PLOS ONE. Congratulations! Your manuscript is now with our production department. 

With kind regards,

on behalf of

Dr. Deborah Dean 

Academic Editor

PLOS ONE